# Inhibition of Kinase Activity and In Vitro Downregulation of the Protein Kinases in Lung Cancer and Cervical Cancer Cell Lines and the Identified Known Anticancer Compounds of *Ziziphus mucronata*

**DOI:** 10.3390/plants14030395

**Published:** 2025-01-28

**Authors:** Themba Sambo, Emelinah Mathe, Leswheni Shai, Sipho Mapfumari, Stanley Gololo

**Affiliations:** 1Department of Biochemistry & Biotechnology, School of Science and Technology, Sefako Makgatho Health Sciences University, Ga-Rankuwa 0208, South Africa; emelinah.mathe@smu.ac.za (E.M.); stanley.gololo@smu.ac.za (S.G.); 2Department of Biochemical Sciences, Tshwane University of Technology, Pretoria 0183, South Africa; shailj@tut.ac.za; 3Department of Physiology, School of Medicine, Sefako Makgatho Health Sciences University, Ga-Rankuwa 0208, South Africa

**Keywords:** tumour, gas chromatography, A549, HeLa, phosphorylation and enzymatic activity

## Abstract

Plants have long been used as sources of natural compounds with therapeutic benefits, providing molecules capable of inhibiting multiple kinases. Many medicinal plants are recognized for their anticancer properties and may offer ways to mitigate the adverse effects of conventional cancer treatments. In this study, the potential of *Ziziphus mucronata* methanol extract as a kinase inhibitor was assessed using the MTT assay, a universal kinase assay, and a human phosphokinase antibody array, along with a GC-MS analysis of volatile anticancer compounds. The MTT assay revealed strong cytotoxicity in A549 cells, with an IC_50_ of 31.25 µg/mL, while HeLa cells showed weaker cytotoxicity with an IC_50_ of 125 µg/mL. In comparison, paclitaxel exhibited potent inhibitory effects on A549 cells (IC_50_ of 31.25 µg/mL) and moderate inhibition on HeLa cells (IC_50_ of 65 µg/mL). Enzyme activity, measured by ADP production in the ADP-Glo assay, indicated that the extract inhibited protein kinase activity in both A549 and HeLa cells after 24 h of treatment. Additionally, the human phosphokinase antibody array, which includes 44 pre-spotted kinases, showed that the extract downregulated multiple phosphorylated kinases in both cell lines. Some of the affected kinases, such as TOR, Fyn, HcK, Fgr, STAT5b, PLC-γ1, p38α, ERK1/2, AMPKA, Akt1/2, GSK-3α/β, MSK1/2, CREB, RSK1/2/3, PLC-γ1, and STAT5a are critical regulators of various cellular processes, including apoptosis, differentiation, and proliferation. The findings of this study suggest that extract from *Z. mucronata* may have the capacity to regulate protein kinase activity, highlighting their significant potential as growth inhibitors for cancer cells.

## 1. Introduction

Cancer is continuously being reported to be one of the leading cause of death worldwide, with over 9.7 million deaths reported annually [1]. This is in spite of developments in the tools of diagnosis, treatment and prevention measures [2]. The number of cases is constantly increasing, and is estimated to be 26 million by 2030 and 17 million cancer-related deaths per year [3,4]. Cancer is a term that applies to a group of diseases that originate from uncontrolled proliferation of normal cells, which produces genetic instabilities and alterations. It accumulates within cells and tissues and transforms a normal cell into malignant cell [5,6]. Some major players in the pathogenesis of cancer development are protein kinases, which are a large family of enzymes responsible for catalysing protein phosphorylation [7,8].

The human genome is reported to have over 500 protein kinase genes [9] that govern a variety of biological activities, including proliferation, cell cycle, apoptosis, and differentiation [10]. Kinase enzymes have been reported to play a crucial role in cancer progression [7,11], as their activity significantly influences these processes [7]. Consequently, the dysregulation of kinase activity can cause dramatic changes, which can be critical for the survival and proliferation of cancer cells [11]. Kinase dysregulation has been linked to a variety of human ailments, including cancer, neurological and infectious diseases [12]. As a result, numerous kinases, such Akt, CDKS, p38, ERBB2, MAPKs, and others, are being explored as potential pharmacological targets [13,14,15,16]. Kinase inhibitors currently account for one-quarter of all current medication discovery and development activities [15]. Over 120 kinase inhibitors have been approved worldwide, largely to treat cancer and autoimmune illnesses [17]. Kinase inhibitors used in tumour treatment are classified into two types: small-molecule kinase inhibitors (SMKIs) and large molecules such monoclonal antibodies (mAbs), nanobodies, and peptides. The EGFR/HER family is the most mature kinase target to date, with 18 inhibitors discovered, including mAbs and derivatives [17]. The most recent SMKIs targeting the EGFR/HER family are largely intended to overcome resistance produced by mutations such as EGFR L858R, T790M, and C797S [18].

The recognition of abnormal protein kinase activity in cancer has significantly transformed the approaches employed in the treatment of cancer [15,19]. Kinase inhibitors, often known as targeted medications, are designed to specifically target and block the activity of uncontrolled protein kinases [20]. Compared to standard chemotherapy, these medications have the advantages of increased specificity, less harm to non-malignant cells, and fewer side effects [21,22]. Examples of such drugs are those like gefitinib and erlotinib, which have shown effectiveness in targeting EGFR mutations associated with certain types of lung cancer [23].

Plants have historically served as rich sources of medicinal compounds, each with unique chemical characteristics [24,25,26,27]. These medicinal compounds are constantly researched for their varied biological effects and this has led to the discovery of active ingredients such as morphine, salicin, emetine, strychnine, colchicine, caffeine, and nicotine [28,29,30]. Advancements in understanding these compounds’ molecular structures have enabled the synthetic production of desired substances, reducing the reliance on natural extraction [31]. Notably, polyphenols are recognized for their antioxidant properties, which include the inhibition of protein kinases by interacting with the ATP-binding site [32,33]. Various flavonoids have even been crystallized in complex with the protein kinase CK2α, showcasing their potential in kinase inhibition [34]. However, it is important to conduct additional studies to identify and authenticate these molecules from medicinal plants which are claimed to have anticancer properties for potential future clinical applications. Amongst the plants used traditionally in South Africa to manage cancer is *Ziziphus mucronata*, which is a botanical species that holds significant importance within the traditional medicinal practices of numerous African groups, due to its recognized medicinal properties [35]. *Ziziphus mucronata*, commonly known as buffalo thorn, belongs to the Rhamnaceae plant family and is mainly found in tropical regions. It is typically found in Southern Africa, along moist riverbanks, and in a limited number of locations in Southeast Africa [36,37]. In a study conducted by Mongalo et al. [37], they found the leaves of *Z. mucronata* to have generous amounts of total phenolic content, total flavonoids, quercetin, terpenoids, quinones and alkaloids when extracted with ethyl acetate and methanol [38]. Additionally, the leaves and roots have been used traditionally to treat various ailments including diarrhoea, tumours, coughs, chest complaints, dysentery, sores, glandular swellings, skin diseases, open and swollen wounds, cancer, ear inflammation, asthma, syphilis, gonorrhoea and measles, as well as rheumatic pains and fever [37,39]. The exploration of new plant compounds with specific properties to inhibit protein kinases is a promising approach for advances in cancer treatment [34,40]. Thus, this manuscript reports on the therapeutic potential of methanol leaf extracts from *Ziziphus mucronata* on the modulation of protein kinases.

## 2. Results

### 2.1. GC-MS Analysis

Extraction of the plant material afforded a yield amount of 2.97 g dried methanol extract, which equals approximately 6%. The recorded extraction yield agrees with observations in the literature, whereby extraction of plant material with methanol gives relatively satisfactory extraction yield [41]. The resultant methanol extract was subjected to GC-MS analysis for detection of compounds, and the results are shown in Table 1.

A total of 25 compounds were identified through GC-MS analysis of methanol extract. Table 1 summarises the bioactive compounds, along with their retention time (RT), molecular formula and molecular weight (MW). The GC-MS analysis of the methanol extract of Ziziphus mucronata leaves revealed the presence of several bioactive compounds: Benzofuran, 7-Methyl-Z-tetradecen-1-ol acetate, 1-Heptatriacotanol, Ethyl iso-allocholate, Spiro[4.5]decan-7-one, 1,8-dimethyl-8,9-epoxy-4-isopropyl, Cryptofauronol, 17-Pentatriacontene, N-(thiazol-2-yl)cinnamamide, Ethyl spiro[2.3]hexane-1-carboxylate, 5-Hydroxymethylfurfural, 2-Methyl-9-β-d-ribofuranosylhypoxanthine, 3-Deoxy-d-mannoic lactone, Acetic acid, 2, propyltetrahydropyran-3-yl ester, 4-Methyloctanoic acid, 2-Vinyl-9-[β-d-ribofuranosyl]hypoxanthine, 2-Pentadecanone, 6,10,14-trimethyl, Hexadecanoic acid, methyl ester, Hexanoic acid, pentadecyl ester, E,E,Z-1,3,12-Nonadecatriene-5,14-diol, 2H-Pyran-2-one, tetrahydro-6-undecyl, 2-(3-Hydroxy-2-pentylcyclopentyl)acetohydrazide, α-Tocospiro A, Viridiflorol, Cycloionone, 6,9,12,15-Docosatetraenoic acid, methyl ester. The molecular structure for some of the detected molecules are presented in the figure below, Figure 1.

### 2.2. In Vitro Cytotoxicity

The cytotoxic effect of Z.mucronata extract was evaluated with an antiproliferative assay on A459 and Hela cell lines. Figure 1 shows the cytotoxic activity of methanol extract against paclitaxel, a common drug used in the treatment of cancer.

Cell viability for the methanol extract at 1000 μg/mL across both HeLa cell lines ranged between 70% and 80%, slightly lower than the untreated cells that ranged between 80 and 90%. The methanol extract was cytotoxic on A549 cells (IC_50_ value of 31.25 µg/mL) and non-cytotoxic on HeLa cells (IC_50_ value 125 µg/mL). Paclitaxel was found cytotoxic on A549 (IC_50_ values 31.25 µg/mL) and moderately cytotoxic on HeLa (IC_50_ values 65 µg/mL). The plant also demonstrated a dose-dependent effect on cell viability in all the cell lines. Methanol extract and paclitaxel demonstrated a stronger antiproliferative activity against A549 cells than the HeLa cells.

### 2.3. Protein Kinase Activity of A549 and HeLa Cell Lines

The impact of Z. mucronata extracts on protein kinase activity was evaluated in A549 and HeLa cancer cell lines, using a universal kinase activity assay. The results, as shown in Figure 2, demonstrate the effects of various treatments on kinase activity in both cell lines.

The graphs in Figure 3 indicate a marked inhibition of kinase activity in both A549 and HeLa cells following treatment with methanol extract and paclitaxel, compared to untreated control cells. Notably, the methanol extract exhibited a stronger inhibitory effect on kinase activity in A549 cells, with a somewhat lesser but still significant inhibition observed in HeLa cells. In contrast, paclitaxel treatment led to relatively weaker inhibition in both A549 and HeLa cell lines, compared to the methanol extract.

### 2.4. Effect of Methanol Extract on the A549 and HeLa Kinase Profile

The effect of the methanol extract of Z. mucronata on protein kinase activity in A549 and HeLa cell lines was assessed using a human phospho-kinase array kit (ARY003B).

The results in Figure 4 showed that the methanol extract downregulated the activity of eight kinases in A549 cells treated with methanol extract, including TOR, Fyn, HcK, Fgr, STAT5b, PLC-y1, p38α, ERK1/2, AMPKA, Akt1/2, and GSK-3α/β. In contrast, five different kinases, MSK1/2, CREB, RSK1/2/3, PLC-Y-1, and STAT5a were downregulated in HeLa cells following treatment with the methanol extract. A549 and HeLa cell lines showed different kinase downregulation profiles in response to paclitaxel treatment. Paclitaxel specifically caused the downregulation of a number of kinases in A549 cells, including ERK1/2, Akt1/2, GSK-3α/β, CREB, YES, PLC-γ1, and HSP60. On the other hand, six kinases: Akt1/2/3, TOR, Fyn, Hck, ERK1/2, YES, PLC-γ, and STAT5a were downregulated in HeLa cells. Remarkably, kinase deregulation was also brought about by the methanol extract treatment: ERK1/2 and PLC-γ1 were downregulated in both A549 and HeLa cells. Notably, ERK1/2, Akt1/2/3, and YES were all deregulated in both cell lines after paclitaxel treatment as can be seen on Figure 5 below. Additionally, the kinase profile of HeLa cells treated with paclitaxel and A549 cells treated with the methanol extract showed downregulation of the kinases Akt1/2/3, TOR, Fyn, and Hck.

## 3. Discussion

Plants have been regarded traditionally as safe and viable alternatives for the treatment, as well as the management, of various ailments and diseases, and it is therefore of paramount importance to scientifically confirm and/or validate the claims. The reported phytochemical constituents of *Ziziphus mucronata* include a diverse array of bioactive compounds, such as flavonoids, alkaloids, terpenoids, and phenolic acids, which have been shown in previous studies [82] to have the ability to interfere with hyperactive tyrosine and serine/threonine kinase signalling pathways implicated in the majority of cancers [83]. In this study, and with GC-MS-based phytochemical analysis, phenolic compounds such as Benzofuran, Spirohexane-1-carboxylic acid, ethyl ester, and 5-Hydroxymethylfurfural were detected in the methanol extract of *Z. mucronata*. Phenolic compounds have been linked to a variety of physiological activities, including cytotoxic effects on different cancer cells [84]. Another class of phytochemicals that is significant for cancer studies are flavonoids, which are known to inhibit the phosphorylation of key kinases like ERK1/2 and AKT, which are critical for tumour cell survival and proliferation [85]. In this study, Cryptofauronol and 2-Methyl-9-β-D-ribofuranosylhypoxanthine, which are known alkaloids, were detected, signifying significant observation regarding claims made concerning *Z. mucronata* about its role in cancer management and treatment. The presence of the identified compounds or their derivatives that are known to possess anticancer properties is an encouraging outcome, as it scientifically positions this plant as a potential source of lead compounds in the fight against the devastating consequences of cancer.

The results of the current study showed possession of cytotoxicity against both the A549 and HeLa cells by the methanol extract of *Z. mucronata* leaves, which was quantified through the IC_50_ determination. It is widely accepted that plant extracts are considered to have promising cytotoxic activity if their IC_50_ is between 30 and 40 µg/mL [86]. The methanol crude extract of this study appears to be outside this range, as the IC_50_ was recorded to be 45 µg/mL after 48 h. This does not, however, imply that it was not cytotoxic, as the IC_50_ of 45 µg/mL can be associated with moderate cytotoxicity. The MTT experiment results show that a 48 h incubation with plant extracts is moderately harmful to cells; however, a longer incubation time may boost cytotoxicity levels. Even though the IC_50_ of *Z. mucronata* methanol crude extract indicates moderate cytotoxicity, it demonstrates the potential for target cell viability in a variety of cancer cells. The observed cytotoxicity activity is supported by the nature of some of the present GC-MS identified compounds within the plant extract, which included phenolic compounds.

Given that various agents, when introduced into the body, interact and affect the function of enzymes including important enzymes like kinases, which are also involved in cancer cell propagation [87], investigating the effect of *Z. mucronata* methanol extract on kinases in this study provided a meaningful and significant contribution to medicinal-plant cancer studies. The results obtained in this study showed the ability of the methanol extract of *Z. mucronata* in the selective downregulation of some protein kinases in both the A549 and HeLa cancer cell lines. Protein kinases are reported to regulate a wide range of cellular functions, many of which are linked to neoplastic changes in cells [15]. Kinase enzymes have also been shown to play a crucial role in cancer progression, and thus are important targets for development of anticancer agents [88]. Kinases act as cell signal modulators, by transferring phosphate groups from ATP to specific substrates, including proteins, carbohydrates, and lipids. Many kinases, including PI3K, serine/threonine kinase, and MAPK, have an effect on these cellular processes [88,89]. The activated signalling proteins included members of the RTK receptor family, which initiate the MAPK pathway. This pathway leads to the phosphorylation of ERKs, JNKs, and p38, which are reported to be very important members of the MAPK family [90,91]. MAPK cascades are essential signalling networks that regulate various cellular functions, including stress responses, apoptosis, differentiation, and proliferation. GC-MS analysis included, amongst others, the detection and identification of alkaloid-type compounds, which included Cryptofauronol and 2-Methyl-9-β-D-ribofuranosylhypoxanthine. Alkaloids are reported to alter the MAPK signalling pathway, which includes the extracellular signal-regulated kinase (ERK), c-jun N-terminal kinase (JNK) and the p38 kinase [83]. As such, the detected alkaloid-type compounds within the methanol extract of *Z. mucronata* could be a contributory factor towards the observed effects on several protein kinases in this study. Therefore, the findings of this study, in that the methanol extract of this plant is able to downregulate these class of kinases, serves as potential confirmation to the claims of the usage of the plant in the management of cancer. The findings of this study further concur with previous studies carried out in different settings, which reported that *Z. mucronata* possesses therapeutic qualities such as protein kinase specificity, cytostatic action, and anticancer properties [92,93]. The outcome of this study lays strong groundwork for the consideration of the signalling protein as a key biomarker for future studies in relation to medicinal plants. This is because phosphorylation/dephosphorylation of protein kinases such as ERKs, CREB, RSK, HCK, and Yes, which are involved in controlling multiple oncogenic pathways and cellular functions like cell growth, proliferation, and survival, provides valuable target points for the development of anticancer agents [13].

## 4. Materials and Methods

Leaves of *Ziziphus mucronata* were collected from Mentz, Ga-Mamabolo in Polokwane, South Africa (−23.8818″ latitude, 29.7521″ longitude). The plant was handpicked by an indigenous knowledge practitioner, and was identified by its vernacular name as moonaona. A voucher number (ZZM76) was allocated, prior to depositing the sample in the herbarium of the School of Science and Technology of Sefako Makgatho Health Sciences University. The leaves were dried at room temperature and later ground to fine powder, using a grinder (Polymix PX-MEC 90 D, Kinematic AG., Luzern, Switzerland), and carefully stored in a cool dry place, awaiting use. A 50 g sample of dried plant powder was soaked in 250 mL of methanol for 72 h on a rocking platform, the homogenate was filtered using a Whatmann number 1 filter paper (Boeco Qualitative filter, grade 3 hw 125 mm, Hamburg, Germany), and the filtrate was evaporated using a rotary evaporator (Eins-Sci Laboratory Equipment, Johannesburg, South Africa). It was then dried under a stream of air until the methanol evaporated, and was stored at room temperature away from light until use.

### 4.1. Gas Chromatography–Mass Spectrometry Analysis

GC-MS analysis of the plant extract of *Ziziphus mucronata* was performed using the Shimadzu’s GC-MS QP2010 series (Shimadzu, Honeydew Roodepoort, Johannesburg, South Africa) with electron impact ionization mode, and a Bpx5 GC column (length, 30 m; thickness, 0.25 m; diameter, 0.25 mm) was used to analyse the samples. Helium gas was used as carrier gas (99.999%) at a constant flow rate of 1 mL/min and an injection volume of 1 μL, in size ratio of 10:1. The ion source temperature was 200 °C, while the injector temperature was 250 °C. The oven temperature progressed from 60 °C, was maintained for 1 min at 300 °C, and stayed elevated for 30 min at a rate of 15 °C/min. The solvent delay was adjusted from 0 to 45 min, and the mass spectrophotometer was configured in positive electron-ionization mode with an ionization energy of 70 eV. A scan interval of fragments from *m*/*z* 35 to 500 Dm was fragmented. The peak area/total peak area ratio was used to compute the relative percentage of each component. Lab Solution was the program utilized, while NIST Coin 4.0 (National Institute of Standards Technology) served as the library. Compounds were identified by direct comparison of their retention indices with MS data from the NIST library (computer library), attached to the GC-MS instrument.

### 4.2. Cell Culture

Human lung carcinoma (A549) and human cervical adenocarcinoma (HeLa) cell lines were obtained from Tshwane University of Technology, Department of Biomedical Sciences, Arcadia Campus. Cells were cultured in DMEM-1640 (Cytiva drug research and manufacturing, Marlborough, England) media, supplemented with 10% foetal bovine serum (FBS) (Capricorn scientific, Ebsdorfergrund, Germany) and 1% antibiotic solution (100 U/mL penicillin, 100 U/mL streptomycin—Invitrogen, Randburg, South Africa) at 37 degrees Celsius in a humidified atmosphere containing 5% CO_2_.

### 4.3. MTT Assay

The effect of methanol extract of the plant on the viability of cells was determined using an MTT assay [94] (Merck Life Science (Pty) Ltd., Modderfontein, Johannesburg, South Africa). A concentration of 1 × 10^4^ cells/well was seeded into a 96 well plate for 38 h, with each well containing approximately 100 μL of cell suspension. After 38 h of incubation, the medium was replaced with 100 μL of extracts diluted to achieve concentrations of 1000, 100, 10, 1, 0.1, and 0.01 μg/mL, and paclitaxel and untreated cells were used as controls. After 38 h, the supernatant was removed and 50 µL of 5 mg/mL MTT reagent was added to each well, followed by further incubation for 2 h at 37 °C, after which, 100 µL of isopropanol was added and it was placed on a shaker, to solubilize the formation of purple crystal formazan. Absorbance was read at 570 nm using an Elisa plate reader (SpectraMax^®^ iD3 multi-Mode). The percentage of inhibition was calculated as previously reported by [95]:$$\%\;cell\;survival\;rate=\frac{average\;Test\;sample-average\;Blank\;}{average\;Control-average\;Blank\;\;}\times 100$$

### 4.4. ADP-Glo Detection-Based Kinase Assays

To evaluate the effect of *Ziziphus mucronata* extract on enzyme activity in A549 and HeLa cells using an ADP-Glo detection assay (Thermo Fisher Scientific, Johannesburg, South Africa), we started by culturing A549 and HeLa cells in a 96-well plate to approximately 80% confluence. The cells were treated with different concentrations of *Ziziphus mucronata* extract in serum-free medium and incubated for 48 h to allow interaction with cellular enzymes. A reaction mixture containing ATP at an optimized concentration to support the targeted enzyme activity was prepared. Following incubation, the ATP-containing reaction mixture was added to each well to initiate kinase or other enzyme reactions, allowing ADP to accumulate in response to enzyme activity. After 30 min, the ADP-Glo reagent was added to terminate the reaction and deplete unconsumed ATP, converting ADP into ATP. After this, a detection reagent was added, which produces luminescence in proportion to the amount of ADP generated, thus reflecting enzyme activity. luminescence was measured using a microplate reader. To determine any inhibitory or stimulatory effect of *Ziziphus mucronata* on enzyme activity, the luminescence values of treated samples and untreated controls were compared.

### 4.5. Human Phosphokinase Antibody Assay

ARY003B (R&D Systems, Barton Abingdon science park, Abingdon, UK) cell lysates were diluted to a total protein concentration and incubated with four membrane arrays pre-coated with capture antibodies, specific for the target proteins. Incubation was carried out overnight at 4 °C to allow both phosphorylated and non-phosphorylated proteins to bind to the membrane. After incubation, streptavidin-HRP was added to the membrane arrays. This secondary antibody binds to the previously added phospho-specific primary antibody, and the membranes were incubated for 1 h, followed by the removal of unbound secondary antibodies. Each membrane was placed on a plastic sheet and uniformly treated with the chemo-reagent mixture. The phosphorylated kinases associated with activated receptors were detected by adding streptavidin-conjugated horseradish peroxidase and a chemiluminescent substrate. The resulting signal was visualized using the Bio-Rad Chemidoc MP Imaging System (Bio-Rad Laboratories Ltd., Johannesburg, South Africa), and quantification was performed by measuring signal intensity (signal intensity O.D/mm^2^) using Bio-Rad software, image lab 5.2.1 [96].

## 5. Conclusions

The current study demonstrated the GC-MS-based phytochemical analysis of the methanol extract of the leaves of *Ziziphus mucronate,* which included the presence of compounds with known anticancer properties. The methanol leaf extract of *Ziziphus mucronata* showed cytotoxic effects against A549 and HeLa cancer cell lines, which highlights this plant’s potential anticancer properties. Finally, the *Ziziphus mucronata* extract demonstrated an in vitro downregulation of specific kinases involved in critical cancer signalling pathways. The findings thus position *Ziziphus mucronata* as a candidate to serve as a source for anticancer agents and cancer-implicated protein kinase inhibitory agents. Further in vivo studies to ascertain the findings of this study are recommended.

## Figures and Tables

**Figure 1 plants-14-00395-f001:**
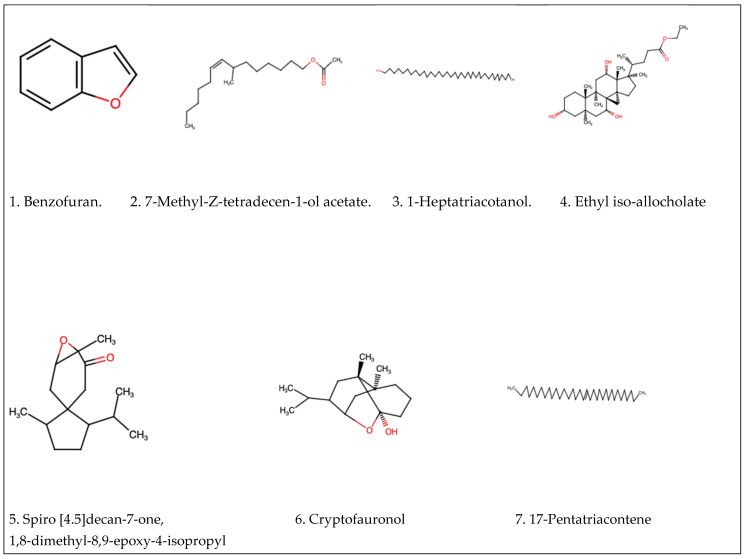
Molecular structures of some anticancer compounds detected by the GS-MS.

**Figure 2 plants-14-00395-f002:**
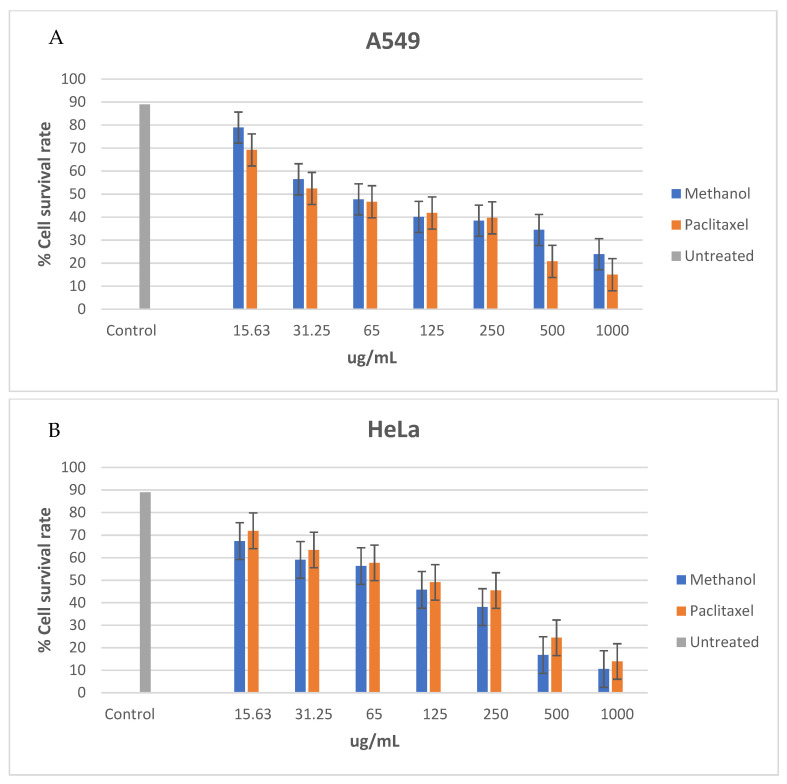
Cell survival rate after 48 h of (**A**) A549 cells treated with methanol leaf extract against paclitaxel and (**B**) HeLa cells treated with methanol extract against paclitaxel; cell survival rate after exposure to methanol extract and paclitaxel for 48 hrs and untreated cells represented as control.

**Figure 3 plants-14-00395-f003:**
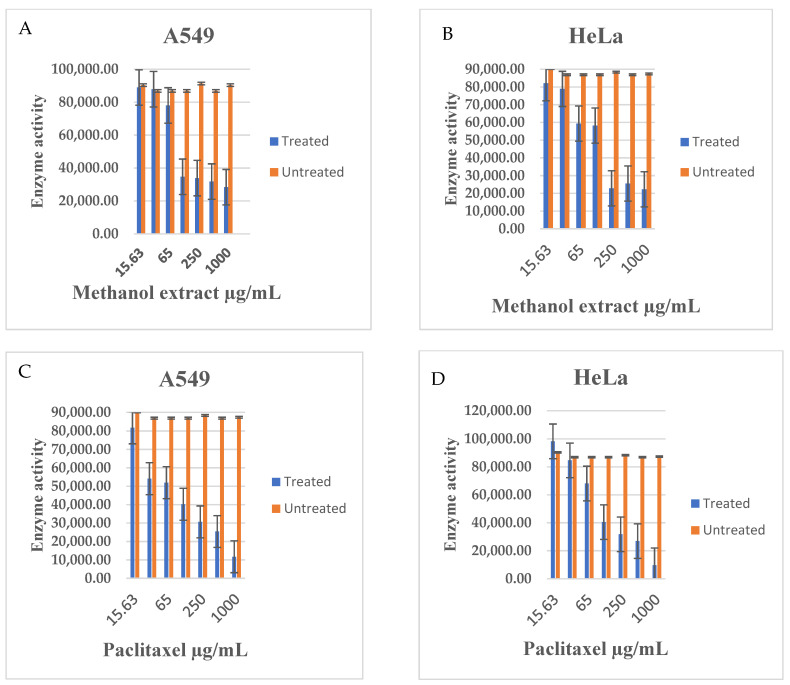
Kinase activity was determined by measuring ATP consumption in relation to ADP using the Universal Glo assay in cell lysates under different treatment conditions after 48 h of exposure: (**A**) A549 cells treated with methanol extract compared to untreated lysates, (**B**) HeLa cells treated with methanol extract compared to untreated lysates, (**C**) A549 cells treated with paclitaxel compared to untreated lysates, and (**D**) HeLa cells treated with paclitaxel compared to untreated lysates.

**Figure 4 plants-14-00395-f004:**
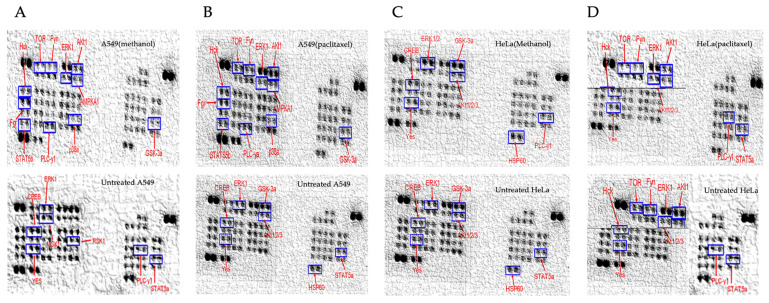
The study investigated the kinase inhibitory profile of *Ziziphus mucronata* methanol extract on A549 and HeLa cells. The cell lysates were analysed for the phosphorylation levels of intracellular kinases using an antibody array (reference). The activation states of the protein kinases, as shown in Figure 4, were assessed. Lysates from A549 cells treated and untreated with the methanol extract and paclitaxel are labelled (**A**) and (**B**), respectively, while lysates from HeLa cells treated and untreated with the methanol extract and paclitaxel are labelled (**C**) and (**D**), respectively.

**Figure 5 plants-14-00395-f005:**
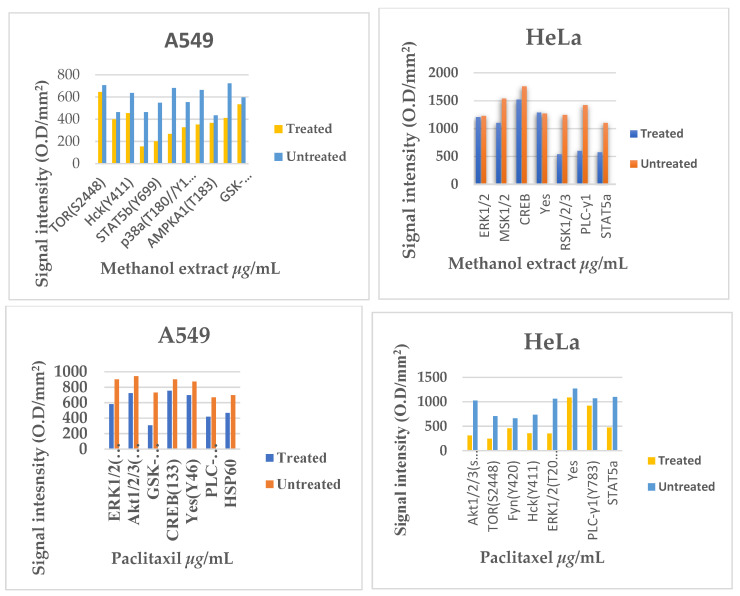
The bar graph represents the expression levels of activated protein kinases of treated cell lysates against untreated cell lysates of A549 and HeLa.

**Table 1 plants-14-00395-t001:** Bioactive compounds found in methanol extract of *Z. mucronata*.

**No.**	**RT (min)**	**Name of the Compound**	**Molecular Formula**	**Molecular Weight (g/mol)**	**Pharmacological** **Actions**
1	04:60	Benzofuran	C_8_H_6_O	118.13	Anticancer activity, anti-tumour, antimicrobial [42,43,44]
2	11:63	7-Methyl-Z-tetradecen-1-ol acetate	C_17_H_32_O_2_	268.4	Anticancer, anti-inflammatory [45]
3	18:14	1-Heptatriacotanol	C_37_H_76_O	536.0	Antioxidant, anticancer, Anti-inflammatory, antimicrobial [46]
4	09:87	Ethyl iso-allocholate	C_26_H_44_O_5_	436.6	Anti-tumour, anticancer, antioxidant [47,48]
5	26:26	Spiro[4.5]decan-7-one,1,8-dimethyl-8,9-epoxy-4-isopropyl	C_15_H_24_O_2_	236.35	Anti-inflammatory, anticancer, antibacterial, antiarthritic properties [49]
6	24:85	Cryptofauronol	C_15_H_26_O_2_	238.37	Antibacterial, antioxidant, cytotoxic activity, anticancer [50,51]
7	26:26	17-Pentatriacontene	C_35_H_70_	490.9	Antioxidant, inti-inflammatory, anticancer [52,53]
8	05:45	N-(thiazol-2-yl)cinnamamide	C_12_H_10_N_2_OS	230.29	Anti-tumour, anti-proliferation, antimicrobial, cytotoxic [54,55,56]
9	06:22	Ethyl spiro[2.3]hexane-1-carboxylate	C_9_H_14_O_2_	154.21	Inhibits KRAS Activity, anti-proliferation [57]
10	06:38	5-Hydroxymethylfurfural	C_6_H_6_O_3_	126.11	Anticancer [58,59]
11	06:56	2-Methyl-9-β-d-ribofuranosyl]hypoxanthine	C_11_H_14_N_4_O_5_	282.26	Anticancer
12	07:37	3-Deoxy-d-mannoic lactone	C_6_H_10_O_5_	162.14	Antioxidant, cytotoxic [60]
13	06:95	Acetic acid, 2-propyltetrahydropyran-3-yl ester	C_10_H_18_O_3_	186.25	Antioxidant, anticancer [61]
14	08:59	4-Methyloctanoic acid	C_9_H_18_O_2_	158.24	Anticancer [62]
15	04:74	2-Vinyl-9-[β-d-ribofuranosyl]hypoxanthine	C_12_H_14_N_4_O_5_	294.26	Anticancer activity [63]
16	12:01	2-Pentadecanone, 6,10,14-trimethyl	C_18_H_36_O	268.47	Antibacterial, anti-inflammatory anticancer [64,65]
17	13:55	Hexadecanoic acid, methyl ester	C_17_H_34_O_2_	270.5	Anticancer, cytotoxic, anti-tumour anti-inflammatory, antimicrobial antioxidant [66,67]
18	15:12	Hexanoic acid, pentadecyl ester	C_21_H_42_O_2_	326.63	Antibacterial, anticancer [68,69]
19	16:04	E,E,Z-1,3,12-Nonadecatriene-5,14-diol	C_19_H_34_O_2_	294.5	Anticancer, anti-inflammatory, antioxidant, cytotoxic [70,71]
20	17:98	2H-Pyran-2-one, tetrahydro-6-undecyl	C_16_H_30_O_2_	254.4	Anti-tumour, anticancer [72,73]
21	09:41	2-(3-Hydroxy-2-pentylcyclopentyl)acetohydrazide	C_12_H_24_N_2_O_2_	228.33	Antifungal, anticancer [74]
22	23:95	α-Tocospiro A	C_29_H_50_O_4_	462.7	Cytotoxic, antiproliferation, anticancer [75,76]
23	26:06	Viridiflorol	C_15_H_26_O	222.37	Cytotoxic, anticancer [77,78]
24	27:29	Cycloionone	C_13_H_20_O	192.3	Cytotoxic, antimicrobial [79,80]
25	15:45	6,9,12,15-Docosatetraenoic acid, methyl ester	C_23_H_38_O_2_	346.5	Anticancer, antioxidant, antimicrobial [81]

## Data Availability

All data for this study have been included in this manuscript, and any more data can be requested from the corresponding authors.

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
