# Peer review of "Inhibition of Kinase Activity and In Vitro Downregulation of the Protein Kinases in Lung Cancer and Cervical Cancer Cell Lines and the Identified Known Anticancer Compounds of Ziziphus mucronata"

_plants, 2025, doi:10.3390/plants14030395_

Round 1
Reviewer 1 Report
Comments and Suggestions for Authors
Authors send their manuscript entitled: „Inhibition of Kinase Activity and In Vitro Downregulation of 2 the Protein Kinases in Lung Cancer and Cervical Cancer Cell 3 Lines and the Identification of the Responsible Anticancer 4 Compounds of Ziziphus mucronata” for revision to Plants journal.
Manuscript needs some improvement and parts these should be changed are listed below:
1. Page 2, line 85: latic plants name should written italic Z.mucronata; also page 3 line 108, page 5 line 147,
2. Page 2.line 87: Authors mentioned paclitaxel as a common medicine used for cancer treatment, but why Autrors did not choose for example doxorubicin other coomon used drug as a reference in tests?
3. Page 3 Did Authors checked cell survival rate only after 48 hrs or for example after 24h or 72h
4. Can Authors write what kind of biologically active compounds are present in this methanolic leaves extract shortly at the begining of Results section instead of right away 2.1. In Vitro Cytotoxicity? And also a content of each. Both, biologists and chemists who will read article will understand such information
5. Page 4, did Authors checked kinase activtity only after 24h or maybe also after 48h were there some diffferences?
6. Page 4 please resize pictures in Fig 3.
7. Page 5. Line 145: 2.4. Gas Chromatography-Mass Spectrometry (GC-MS) Analysis of Methanol Extract of Ziziphus mucronata thids section should be at the begining od Result section
8. Page 6 Can Authors nimbered each detected compounds in main manuscript text and also in figures
9. Page 6 , line 162 ; what does mena this ; $$ Viridiflorol
10. Page 7, there should be Figure 7 instead do 2.
11. Could Autrors add chemical structures drawn in one way.
12. Experiments were duplicated or triplicated? Please add to the description of respective figures
13. Page 9 How many g of dried extract was obtained
14. Page 9 , line 276; Names should be written in capital letters; Tshwane university of technology, department of biomedical sciences, arcadia campus.
Author Response
Dear reviewer
Thank you for the time you dedicated to reviewing our manuscript. We appreciate your candid feedback and recommendations. We have addressed your comments on the revised submitted document and responded to your comments on the attached document.
Regards

Reviewer 2 Report
Comments and Suggestions for Authors
The paper of Themba sambo, Emelinah Mathe, Leswhene Shai, S.Mapfumari, Stanley Golodo "Inhibition of kinase activity and in vitro downregulation of the protein kinases in lung cancer and cervical cancer lines and the identification of the responsible anticancer compounds of Ziziphus mucronata" is devoted to the study of plant Ziziphus mucrolata leaf methanol extract as the kinases inhibitors playing a crucial role in cancer progression. As the results of the study, it was found that methanol extract of this plant showed cytotoxic effects in A549 cells with IC50 31.25 micromol (analogous to paclitaxel), and Hela cells (IC50 125 micromol) in MTT-assay. Enzyme inhibitory activity showed that the methanol extract inhibited specific protein kinase activity and downregulated multiple phosphorylated kinases involved in critical cancer signaling pathways in both A549 and Hela cells. The GC-MS analysis was carried out and 25 compounds were identified in the methanol extract studied.
The manuscript is useful for researchers in the field of plants phytochemicals, phytomedicine, and development of new cancer cells growth inhibitors based on the available plant's resources, and may be published after corrections according comments given in PDF of manuscript, the main of which are the next:
1) There are no references in the Discussion and Experimental part reflecting the used assays for in vitro studies.
2) Corrections should be made in captions and subtitles, and enclosed table.
3) The plant's extracts should be quantitatively standardized according the certain parameters or content of some bioactive second metabolites because of they may differ depending of used raw material, extraction conditions, etc.
4) Ther are no data about the plant's collection number or the accession number of used plan'r material.
Values p should be indicated in the captions to graphs as p less ....

Author Response

(The authors gave the same response as above.)

Reviewer 3 Report
Comments and Suggestions for Authors
Sambo and colleagues prepared a manuscript on inhibition of kinase activity and in vitro downregulation of the protein kinases in lung cancer and cervical cancer cell lines and the identification of the responsible anticancer compounds of Ziziphus mucronata. As follows from the abstract of the article, the authors assessed the potential of the methanol extract of Ziziphus mucronata as a kinase inhibitor, a universal kinase assay, and an array of antibodies to human phosphokinase. In addition, volatile compounds were analyzed by GC-MS.
The formulation of the manuscript raises many questions, not to mention its content.
1. From the beginning of the manuscript, the authors mislead the readers, since the title does not correspond to the content of the manuscript - the identification of anticancer compounds of Ziziphus mucronata is not discussed in the manuscript itself.
2. Chapter Introduction contains detailed information on various kinases and why they should be studied in the development of antitumor agents. However, not a word is said about the choice of the authors - why they decided to study the methanol extract of Ziziphus mucronata, what kind of plant is it, what is remarkable about it, what is the relevance and novelty of the studies conducted
2. The very arrangement of the chapters in the manuscript raises questions - classically, they first describe the extraction of the extract, identify the compounds it contains, and then move on to describing the study of biological activity. The same logic is followed in the subchapters in the methods chapter. This logical chain should be followed; if any methods are known, the relevant literary sources should be cited.
4. Activity values (IC50, etc.) are usually designated in µM
5. Figures should not contain incomprehensible designations, for example, what does methanol mean? The title of chapter 2.3. Effect of Methanol on the A549 and HeLa Kinase Profile? It is clear that the authors mean the methanol extract of the plant, but such designations are unacceptable. In addition, there are no designations for the figures in the text. Figure 3 is not clear at all what it means.
6. Subchapters 2.1. and 2.2 should also contain literature data on the activity of Paclitaxel
7. Subchapter 2.4. - why are these 7 compounds listed in the table and figure? Where are the rest of the 25 identified? In addition, both the names of the compounds and the figures of the structures of the compounds listed in this chapter contain numerous errors, which indicates the authors' lack of competence in this matter
8. The discussion chapter discusses common known issues, and the discussion of the results of the study is devoted to one small last paragraph, from which I still do not understand how the results of this study highlight the potential of medicinal plants as sources of multikinase inhibitors and what contributes to the overall therapeutic potential of Ziziphus mucronata.
9. The conclusion does not contain specific conclusions based on the results of the studies
The manuscript is poorly written, contains a lot of grammatical and spelling errors and in its current form is not suitable for publication in the journal Plants
Author Response

(The authors gave the same response as above.)

Round 2
Reviewer 1 Report
Comments and Suggestions for Authors
Authors improved their manuscript according to reviewer's suggestions
I have no further remarks.
Author Response
Dear reviewer
Thank you for your feedback, we appreciate your constructive comments and guidance.
Regards
Reviewer 2 Report
Comments and Suggestions for Authors
The paper of Themba Sambo, Emelinah Mathe, Leswhene Shai et al "Inhibition of kinase activity and in vitro downregulation of the protein kinases in lung cancer and cervical cancer cell lines and the identification of the responsible anticancer compounds of Ziziphus mucronata" is corrected according the recomendations and may be accepted for publication in Plants.
Reviewer 2

Author Response

(The authors gave the same response as above.)

Reviewer 3 Report
Comments and Suggestions for Authors
1. Since only the extract was studied for biological activity, and the identified compounds were not studied separately and in comparison with the extract, I still think that the title of the manuscript is incorrect.
2. Some figures contain errors in the names of compounds, the chemical names of the compounds identified in the methanol extract in the text should be written with a small letter, compound 5 in the table should be given one name, compound 13 - 2-Propyltetrahydro-2H-pyran-3-yl acetate, similar comments for compounds 16, 17, 18, 20. And in principle, it would be good for the authors to decide and use either trivial names or IUPAC nomenclature.
3. Figure 1 - the authors decided to cite only some anticancer compounds, they should be accompanied by appropriate literary references
4. The Discussion chapter has changed little after editing and is still very poor and contains minimum information
5. The conclusion remains unchanged.
6. The manuscript still contains numerous typos and errors and in its current form is not suitable for publication.
Author Response
Dear Reviewer
Thank you for your feedback. We have addressed your comments and recommendations in the attached document.
Regards

Round 3
Reviewer 3 Report
Comments and Suggestions for Authors
The authors have improved the manuscript sufficiently and taken into account almost all of my comments.